# Modeling of Probeless Friction Stir Spot Welding of AA2024/AISI304 Steel Lap Joint

**DOI:** 10.3390/ma15228205

**Published:** 2022-11-18

**Authors:** Mariia Rashkovets, Nicola Contuzzi, Giuseppe Casalino

**Affiliations:** Dipartimento di Meccanica, Matematica e Management, Polytechnic University of Bari, Via Orabona 4, 70125 Bari, Italy

**Keywords:** probeless friction stir spot welding, dissimilar weld, numerical analysis, material flow, microstructure

## Abstract

In the present study, AA2024 aluminum alloy and AISI304 stainless steel were welded in a lap joint configuration by Probeless Friction Stir Spot Welding (P-FSSW) with a flat surface tool. A full factorial DOE plan was performed. The effect of the tool force (4900, 7350 N) and rotational speed (500, 1000, 1500, 2000 RPM) was analyzed regarding the microstructure and microhardness study. A two-dimensional arbitrary Eulerian–Lagrangian FEM model was used to clarify the temperature distribution and material flow within the welds. The experimental results for the weld microstructures were used to validate the temperature field of the numerical model. The results showed that the tool rotation speed had an extensive influence on the heat generation, whereas the load force mainly acted on the material flow.

## 1. Introduction

The design of efficient and lightweight products encourages engineers to join dissimilar structures. One of the most promising material couples used in automotive industries is the aluminum–steel joint due to its specific strength and high corrosion resistance [1]. About 90% of the assembly in automotive or aerospace products is performed by spot welding, where Resistance Spot Welding (RSW) is actively used [2]. RSW typically involves compressed metal plates between two electrodes to produce an electrical impulse that heats and melts plates at the contact point, with subsequent crystallization. However, melting forms a layer of brittle intermetallic phases between aluminum and steel [3] or even unexpected phases between the same grade of materials [4,5]. Different modifications of RSW can reduce unwanted phases [6,7]; however, the welding process becomes more complicated and the mismatch in the aluminum–steel bonding remains [8,9].

Friction Stir Spot Welding (FSSW) is a solid-state process that allows for the joining of different materials by a combination of friction and plastic deformation, avoiding the problems that result from their melting and solidification. Spot welding is created by plunging a non-consumable rotating tool consisting of a cylindrical shoulder and a probe into the workpiece with further material stirring. Therefore, the total amount of heat input generated during the friction must be sufficient to soften, stir, and plastically deform materials. Although there are many reports on optimizing FSSW (e.g., [10,11]), the most pronounced keyhole and ‘hooking’ defects owing to the probe remain [12]. Moreover, the wear or failure of a probe made of expensive material (e.g., WC-Co or CBN) can cause defects such as a lack of penetration [13]. All probe-related defects are crucial for the welds’ performance, which also affects the cost of the whole process. For these reasons, several different processes have been proposed.

Recently, the refill FSSW process was applied to eliminate keyholes due to the separate movement of the tool [14]. However, the process required a very complex machine, as well as high control and the regular cleaning of a tool-holder system. Along with these, defects such as voids, hooks, annular grooves, and insufficient refill are still observed [15]. D. Bakavos and P.B. Prangnell [16] produced results that went against common wisdom that the probe must penetrate at least 25% into the bottom plate to make an acceptable weld. Later, Y. Tozaki et.al [17] and other authors [18,19,20,21] demonstrated the possibility of producing quality welds using a probeless tool. The ability of the simple probeless FSSW (P-FSSW) process to create a full metallurgical bond, eliminate keyholes, and improve tool life was shown. 

The movement of the material within the lap welds during friction processes is more important than the microstructure due to the interface between the plates [22]. Nowadays, complex numerical models estimate the material flow in FSSW and P-FSSW [23,24]. The results of various studies have shown variations in the failure modes attributed to the different material flow between the plates. Currently, authors suggest using a probeless tool with a grooved shoulder since it ensures a deeper material flow to the bottom plate and yields higher failure energies [25]. However, grooves filled with softened aluminum create the need for frequent tool cleaning and the excessive material flow can lead to microcracks and voids.

Most of the mentioned studies were based on the welding of light materials such as Al, Li, and Mg by a probeless tool with a grooved shoulder, whereas fewer studies focused on deep aluminum–steel examination. Based on previous results, the P-FSSW process for an Aluminum–steel couple using a probeless tool with a flat shoulder was the focus of this study. The main goal was to clarify the feasibility of a completely flat tool to weld a dissimilar AA2024-AISI304 lap configuration using different process parameters for P-FSSW, as well as the quality of welds. A two-dimensional arbitrary Eulerian–Lagrangian (ALE) FEM model was validated with experimental data to analyze the material flow and temperature distribution. The scope of the experimental and numerical study was the observation of four levels of rotation speed and two levels of tool vertical force.

## 2. Materials and Methods

### 2.1. Materials

Welds were made using commercial rolled aluminum plates of AA2024 (0.8 mm) and AISI304 stainless steel (6 mm) in a lap configuration. The chemical compositions of the used materials are presented in Table 1. The initial microstructure of the commercial rolled AA2024-T3 plate had equiaxed grains with an α-solid solution of Cu-Al (the bright contrast) and secondary phases (the dark contrast), whereas the rolled AISI304 plate was characterized by a typical microstructure with elongated grains in a rolling direction (Figure 1). The thermal properties of the used materials are listed in Table 2.

### 2.2. Methods

Since acceptable friction welding cannot occur over a wide range of plunge depths, the P-FSSW process was performed in a force control mode: (1) as the rotating tool at the set rotation speed began to contact the top surface of the upper plate, the force control mode was activated and the control system monitored the loading force of the tool until the set force parameter was reached; (2) once the force load reached the set value, the tool started to experience micro-up-to-down movements to keep the set parameter constant throughout the P-FSSW process. The lap configuration is presented in Figure 2. The probeless tool was made of H13 steel with a 30 mm shoulder diameter and a tilt angle of 0 degrees. The parameters of the P-FSSW process were chosen based on the preliminary experiments (Table 3). The dwell time was set at 30 s for each specimen. The rotation speed of 500 RPM coupled with the 7350 N and 4900 N load forces did not generate a sufficient heat input to soften the workpiece material and welding was not achieved. Therefore, those samples are overlooked in the following discussion.

Samples for optical microscopy were cut perpendicular to the weld spot by an electrical discharge cutting machine. The aluminum microstructure was etched with Keller’s reagent (1.5 mL HCl, 2.5 mL HNO_3_, 1 mL HF, and 95 mL distilled water), whereas the mixture of 1 mL HNO_3_ and 3 mL HCl was applied to etch the surface of the stainless steel.

### 2.3. Numerical Model

For the prediction of the thermal field and material flow in the P-FSSW process for AA2024/AISI, 304 welds were performed using the finite element software Simufact Forming^®^ 2021. Based on the previous studies [24,26], the metal flow was purely axisymmetric; therefore, quad 2D elements were used to mesh all the parts of the P-FSSW CAD model (Figure 2b). In the model, the rotating tool, clamping system, and substrate were defined as rigid bodies, whereas the plates to be welded were deformable bodies. The initial element sizes were chosen as 0.07 mm for the AA2024 plate (upper plate) and 0.2 mm for the AISI304 plate. The initial element sizes of the rigid bodies varied from 0.2 mm (tool, clamping system) to 0.7 mm (substrate). The friction coefficient applied between the workpiece and the tool governed by Coulomb’s friction law with a simplified contact condition of sliding friction was 0.1.

The change in the contact pressure (P) and tool rotation speed (RS) over time consisted of the following stages:Linear increase in the RS from zero (RS_0_) to the rated value of RS_set_ for separate specimens (P_0_ = 0).Tool downward movement until it touched the plate top surface (RS_set_; P_0_).Fast compression of the aluminum plate with an instant increase in the pressure up to the rated value of P_set_. Frictional heat was generated, which was proportional to the specific frictional power.Formation of welding seam for the DT (RS_set_; P_set_).Tool upward movement, with a pressure decreasing to P_0_ (RS_set_).Linear decrease in the RS from RS_set_ to RS_0_ for separate specimens.

Stages 2–5 were implemented in the presented FEM. An arbitrary Eulerian–Lagrangian finite element method was used for the plates (the mesh moved independently from the material in a way that spanned the material at any point in time), whereas the tool, clamping system, and substrate were calculated as rigid Lagrangian bodies. 

The material flow was simulated as a bi-dimensional fluid flow through an enclosed volume in accordance with the following equations of equilibrium:(1)$$\frac{D\left(\rho v_{i}\right)}{Dt}=\rho b_{i}+\frac{\partial \sigma_{ij}}{\partial x_{j}}$$
where $\frac{D}{Dt}\;$ is the material time derivative of a quantity, *v_i_* is the velocity of the particle flowing through the mesh, $\sigma_{ij}$ is the Truesdell rate of Cauchy stress, ρ is the material density, and *b_i_* is the body force. For an incompressible fluid, Equation (1), along with the continuity equation (mass conservation), yields:(2)$$\rho \frac{\partial v_{i}}{\partial t}+\rho v_{j}\frac{\partial v_{i}}{\partial x_{j}}=\rho b_{i}+\frac{\partial \sigma_{ij}}{\partial x_{j}}$$

The left-hand side of the continuity equation, Equation (2), represents the local rate of change augmented by the convection effects. 

Experimental data such as torque or efficient energy cannot be directly used for the validation of the results according to simplify assumptions of a built-in 2D model. However, a 2D model can predict many trends corresponding to the experimental data such as the material flow. Arrow plots were used to visualize the material flow.

## 3. Results and Discussion

### 3.1. Microstructure

The upper-surface view and corresponding cross-section for each sample are presented in Figure 3. Several conditions were observed among the upper surfaces and corresponding cross-sections: thinning with a small flash volume (AS-01, AS-05), partial distortion with a medium flash volume (AS-02 and AS-06), and a complete metallurgical bond with the highest (AS-03) and least (AS-07) volumes of flash. 

The degree of material softening was affected by the duration of the process at a certain temperature. The higher temperature generated at 1500 RPM coupled with both load forces of 7350 N and 4900 N was responsible for an excessive heat input with a great softening and distortion of the upper AA2024 plate, which also matched the equivalent stress result owing to fast material heating. A combination of the highest rotation speed (2000 RPM) and both cases of tool force produced the thinning of the upper plate in samples AS-01 and AS-05. Detailed images show that there were no refined grains (Figure 4). The samples made at the lowest rotation speed (1000 RPM) had a good metallurgical bonding due to the reduction in the heat input per unit length and lower dissipation of heat over a wider region of the workpieces, which made the heating slow and appropriate. The difference in the tool axial force was responsible for a large gap in the flash volume of samples AS-03 and AS-07. The higher amount of flash in sample AS-03 was due to stronger tool penetration into the workpiece, which was also demonstrated by the numerical results.

Figure 5 show detailed cross-sections of P-FSSP welds at a rotation speed of 1000 RPM under load forces of 7350 N (AS-03) and 4900 N (AS-07), respectively. The effect of severe plastic deformation was seen in the upper part of the AA2024 plate, which experienced elevated temperatures and rapid recrystallization (Figure 5a). 

The middle part of the AA2024 plate had a very narrow zone with non-recrystallized and elongated grains (TMAZ) caused by the material flow and shear stress. The shear stress took place only in the upper and middle parts of the aluminum plate, whereas the material close to the interface between the AA2024 and AISI304 plates did not experience any plastic flow. This zone was influenced only by the heat. Furthermore, a slight thermal cycle did not permit the recrystallization of the aluminum. This was probably due to the lower conductivity of the steel and the presence of the Thermal Contact Conductance (TCC) at the interface, which represented the fraction of heat that was channeled from the aluminum and steel plates. The TCC did not allow for a complete heat exchange between the two plates and a fraction of the heat persisted in the AA2024 plate, decreasing the cooling rate [27].

The AISI304 side showed the rolled grain structure of the initial state (see Figure 1b) without microstructure changes. 

The higher average temperature was accompanied by a longer heating time, which promoted a greater volume of material to be heated and a wider zone with dynamic recrystallization. The corresponding dynamic recrystallization zones were 125 μm (sample AS-03 with T_av_. = 400 °C) and 155 μm (sample AS-07 with T_av_. = 430 °C). The resistance of the material flow with a greater volume of refined grains decreased significantly [28], which explains the differences in the material flow patterns of the numerical and experimental results.

### 3.2. Material Flow

The material underneath the shoulder in both cases was pushed downward with further outward extrusions (Figure 6c,d), which correlates with [26]. Figure 6 represents the stages of the simulated material flow for samples AS-03 and AS-07, namely the start (5–10 s), middle (15 s), and final (30 s) stages and 10 sec after the FSSP process was completed. The main difference in the material flow between the two cases was observed at 10 sec (Figure 6b) when the higher force in sample AS-03 led to the obvious shift of hotter material outside the tool and created flash (Figure 7e,f), whereas the lower tool force in sample AS-07 contributed only to the material flow (Figure 7a–d). The subsequent increase in middle stage temperature improved the material flow in sample AS-03 (Figure 6c) and some similarityi in the material flow remained until the final stage of the P-FSSW process. After the tool was retracted, sample AS-07 had a more uniform return with a high concentration at the center compared to sample AS-03 (Figure 6e).

### 3.3. FEM Analysis: Temperature and Stress

The graphs of the average temperature and equivalent stress along the tool shoulder–workpiece interface during the P-FSSW process for different process parameters are shown in Figure 8. The average temperature for all the samples in the tool shoulder–workpiece interface was about 400 °C, whereas an evident difference in temperature was seen in the samples made using 7350 N of tool vertical force. The equivalent stress decreased at a low rotation speed due to the dissipation of heat over a wider area of the workpiece under slower material heating.

Figure 9 shows the numerical results on the influence of the welding parameters regarding the temperature field in the cross-sections of weld spots. The temperature had a uniform distribution with the maximum value found along the tool shoulder–workpiece interface. The maximum temperature of a friction solid process can range from 80% to 90% of the melting point as measured by Tang W. et al. [29] and the average temperature at the shoulder–workpiece interface can reach 90–95% of the melting temperature [30]. The maximum {peak} temperature of 630–640 °C at the AA2024-AISI304 interface was generated at 2000 RPM (samples AS-01 and AS-05), whereas the minimum temperature of about 610 °C was observed at 1000 RPM (samples AS-03 and AS-07). Both peak temperatures were greater than the melting point of the AA2024 alloy (602 °C) due to the great differences in the materials’ thermal properties (see Table 2) and the thermal contact resistance between the two plates (Figure 9b).

### 3.4. Microhardness

All the samples showed similar results for the average microhardness profiles (Figure 10). Figure 11 presents the microhardness trends in the cross-sections of each P-FSSW weld. The upper area of the aluminum plate was characterized by a higher microhardness with respect to the base metal due to the grain refinement and a reprecipitation process that might have taken place during the cooling of the heat treatable AA2024 alloy, whereas the TMAZ and HAZ were low due to the coarsening or dissolution of the hardening precipitates. The main differences in the microhardness distributions of samples AS-03 and AS-07 were consistent with the material high plastic deformation flow, and the most uniform microhardness distribution was seen in sample AS-07.

## 4. Conclusions

In the current study, commercial rolled plates of AA2024 aluminum alloy (0.8 mm) and AISI304 stainless steel (6 mm) were lap-welded with a Probeless Friction Stir Spot Welding (P-FSSW) process at two levels of axial force and four levels of rotation speed. There were two main goals for this study. The first goal was to perform and investigate the effect of the rotation speed and tool force on the microstructure and microhardness of spot-dissimilar AA2024-AISI304 welds in a lap configuration made with a flat tool. The second goal was to validate the numerical results and clarify the temperature distribution and material flow within the workpiece during P-FSSW with a flat tool.

This study demonstrated the feasibility of the use of a probeless tool with a flat shoulder for solid-state aluminum–steel welding. The micrographic analysis did not reveal intermetallic compounds typical in fusion welding. The results suggest that the rotating tool transports the material only at the upper part of the AA2024 plate with high plastic deformation. The increase in microhardness is accompanied by dynamic recrystallization and spreading of HAZ at depths throughout the entire cross-section of the workpiece. The tool rotation speed has a significant influence on heat generation, which is supported by the numerical results on the temperature distribution, whereas the load force provides the main contribution to the material flow. The numerical results show that a completely different and strong material flow occurs in the first stage of the P-FSSW process. Overall, variations in the average temperature and stress during the P-FSSW process were seen to be relatively low for all welds compared to the variation in the stress of the samples obtained at 1000 RPM and with both load forces. Consequently, it appears that the material flow during the P-FSSW process is primarily driven by the load force. The optimal rotation speed of 1000 RPM, coupled with 4900 N and 7350 N load force, provides sufficient forging pressure and successful welding with a noticeable difference in the material flow. To consider the balance of the tool force at 1000 RPM and to estimate the material flow at the lap joints, further studies are needed to optimize the welding parameters

## Figures and Tables

**Figure 1 materials-15-08205-f001:**
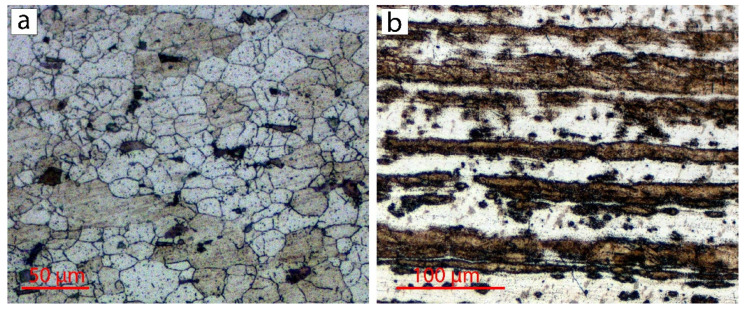
Initial microstructure of AA2024-T3 plate (**a**) and AISI304 plate (**b**).

**Figure 2 materials-15-08205-f002:**
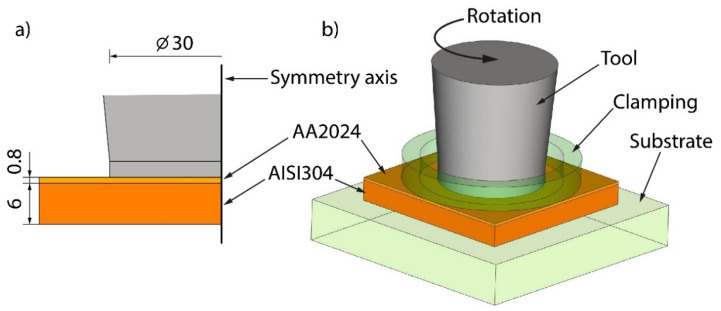
P-FSSW lap configuration: cross-sectional dimensions (**a**), CAD model (**b**).

**Figure 3 materials-15-08205-f003:**
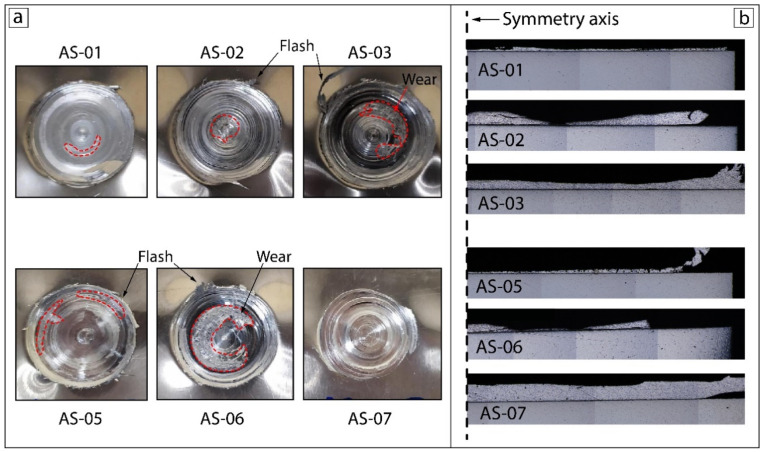
The upper-surface appearances (**a**) and cross-sections (**b**) of P-FSSW samples.

**Figure 4 materials-15-08205-f004:**
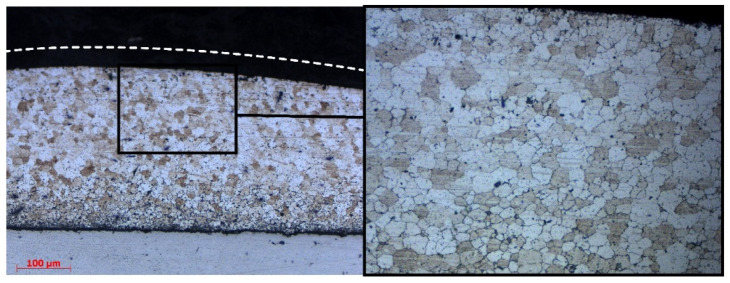
Microstructure of sample AS-01 under a 7350 N load force and at 2000 RPM.

**Figure 5 materials-15-08205-f005:**
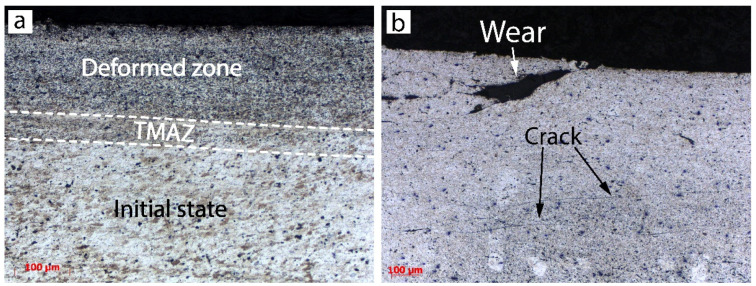
Microstructure in cross-section of sample AS-03: different areas of upper plate (**a**) and defects (**b**).

**Figure 6 materials-15-08205-f006:**
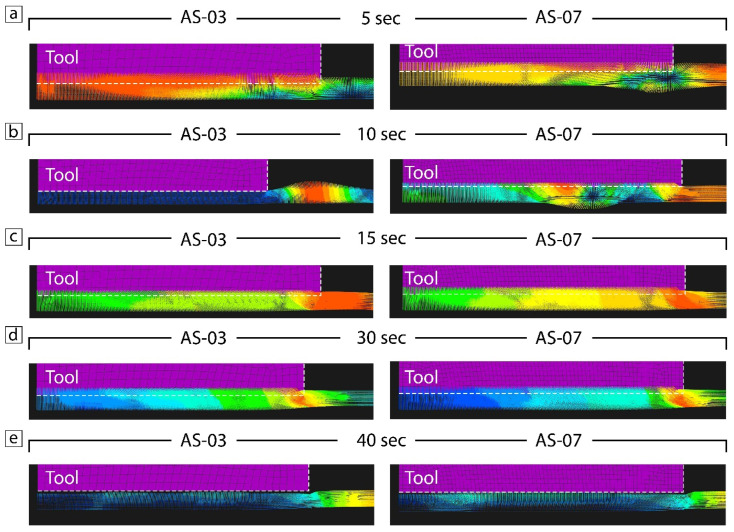
Material flow in samples AS-03 and AS-07 at the different time stages (**a**–**e**).

**Figure 7 materials-15-08205-f007:**
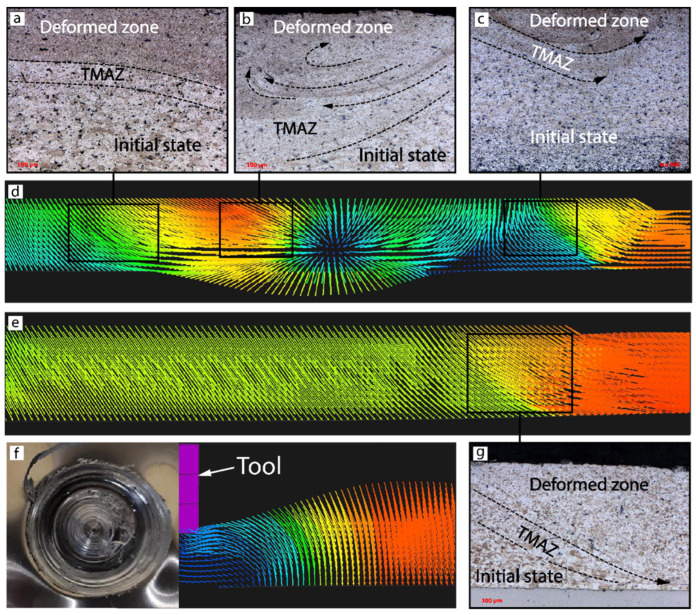
Microstructure (**a**–**c**) and material flow (**d**) of AA2024 plate in sample AS-07 at 1000 RPM and a 4900 N load force; material flow (**e**,**f**) and microstructure (**g**) of AA2024 plate in sample AS-03 at 1000 RPM and a 7350 N load force. The blue color represent low material flow, while the red one high material flow.

**Figure 8 materials-15-08205-f008:**
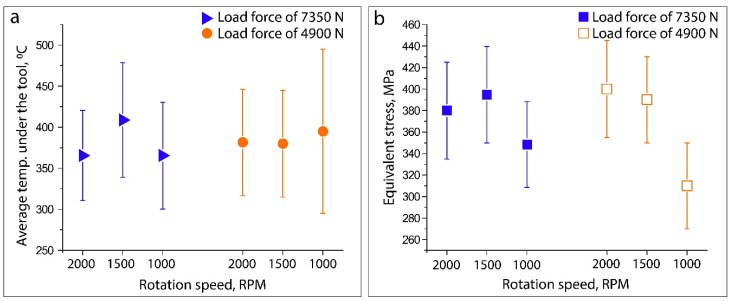
Numerical results of average temperature (**a**) and equivalent stress (**b**) along the tool shoulder–workpiece interface as a function of load force and rotation speed combination.

**Figure 9 materials-15-08205-f009:**
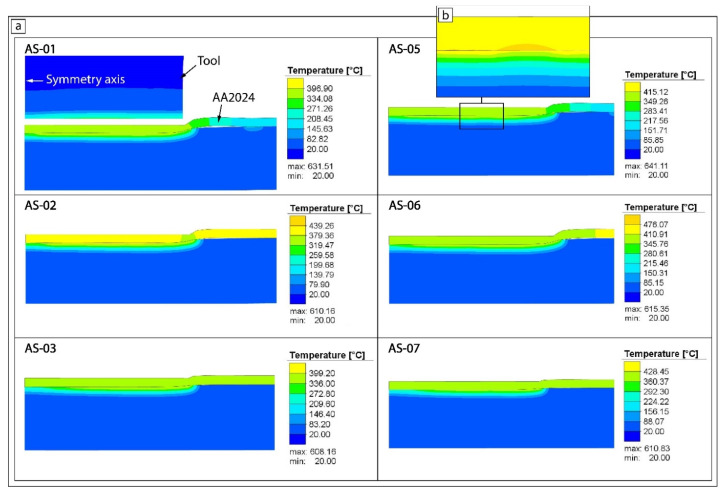
The average temperature distribution profiles (**a**) and peak temperatures on the interface (**b**).

**Figure 10 materials-15-08205-f010:**
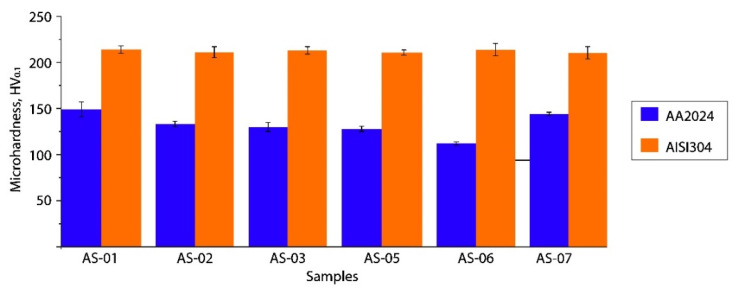
Microhardness distribution.

**Figure 11 materials-15-08205-f011:**
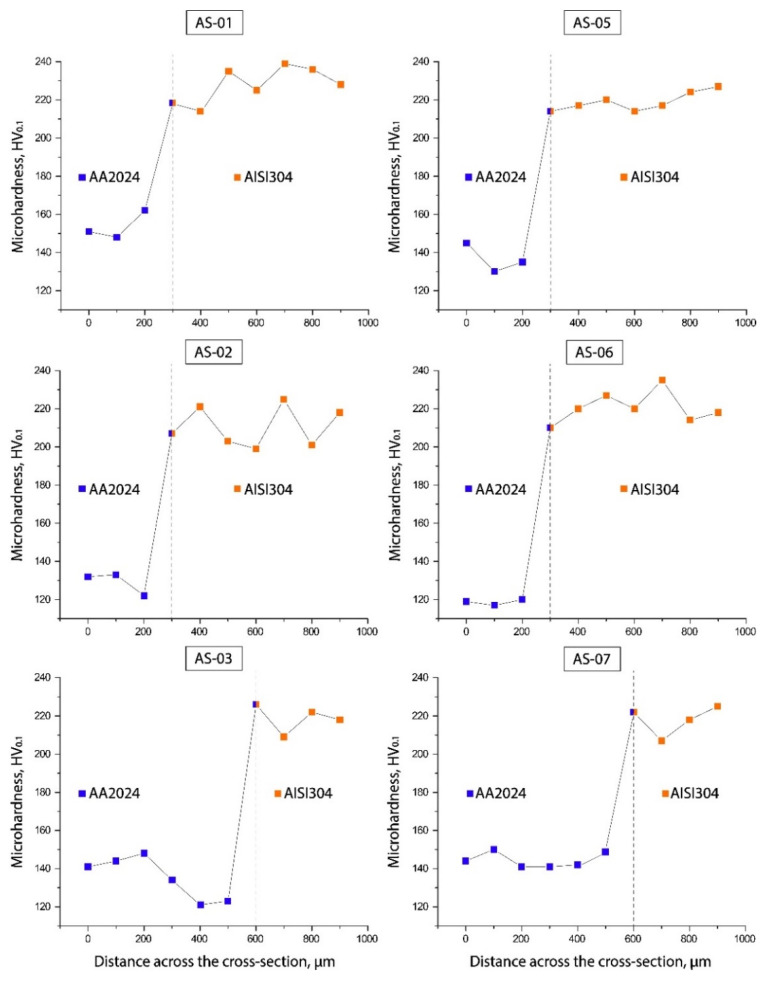
Microhardness profiles in cross-sections of the P-FSSW welds.

**Table 1 materials-15-08205-t001:** Chemical compositions of used materials.

Materials	Elements [wt.%]										
	Fe	Ni	Al	Cu	Mg	Mn	Si	Cr	Zn	Ti	C
AA2024-T3	≤0.5	-	Bal.	3.8–4.9	1.2–1.8	0.3–0.9	≤0.5	≤0.1	≤0.25	≤0.15	
AISI 304 *	Bal.	8–10.5	-	-	2	-	1	17.5–19.5	-	-	0.07

≤0.05% P and ≤0.03% S *.

**Table 2 materials-15-08205-t002:** Thermal properties of used materials.

Materials	Melting Point [°C]	Thermal Conductivity [W/m K]	Specific Heat Capacity [J/g °C]
AA2024-T3	502–638	121	0.875
AISI 304	1400–1455	16.2	0.5

**Table 3 materials-15-08205-t003:** Process parameters.

Test	Rotation Speed [RPM]	Tool Force [N]
AS-01	2000	7350
AS-02	1500	
AS-03	1000	
AS-04	500	
AS-05	2000	4900
AS-06	1500	
AS-07	1000	
AS-08	500	

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
