# Peer review of "Modeling of Probeless Friction Stir Spot Welding of AA2024/AISI304 Steel Lap Joint"

_materials, 2022, doi:10.3390/ma15228205_

Round 1

Reviewer 1 Report

I appreciated the work done by the authors, while the English should be improved before acceptance.

Author Response

The authors thanks the reviewer for the suggestion. The authors have carefully proofread the manuscript and have made the needed corrections.

Reviewer 2 Report

Overall, the paper was well presented. However, the only my concern is on references, where most of them are outdated. Please add more recent published and relevant articles. My suggestion is more 50% of the references are less than 5 years.

Author Response

Thank you for the constructive comment. The references were revised in accordance of reviewer suggestion.

Reviewer 3 Report

Manuscript ID: materials-2014230 entitled “Modelling of Probeless Friction Stir Spot Welding of AA2024/AISI 304 steel lap joint” for journal of Materials has been reviewed.

The subject is potentially interesting and consistent with the data to be proposed to the readers of "Materials".

The Manuscript may be considered after revision, however, some points should be addressed and discussed in the revised version, as mentioned below:

1- The novelty of the study should be further explained (in introduction…).

2-  More references (related to different studies) should be added to the introduction. (recent studies, 2021-2022).

3- In the ıntroduction section;

 it will be useful to give general information about welding technologies.

4- How were the experimental parameters chosen? (according to literature or?)

5- The resolution of Figure 1, 3, 4 and 5 should be increased. (and magnify)

6-  In “Results and discussion” section;

……..On the contrary, the material close to the interface between AA2024 and AISI304 plates did not experience any plastic deformation. The middle part of AA2024 plate had a very narrow zone with non-recrystallized and elongated grains (TMAZ)……….. Why? Explain? (more detailed)

- The graph in Figure 10 should be in more prominent colors.

- The explanations about the hardness values given in Figure 10 should be detailed.

7-  Conclusions section should be enriched a little more.

8- More literature studies should be added to the introduction and other sections (DOIs given below).

DOI-1 https://doi.org/10.1007/s13369-021-06243-w 

DOI-2  https://doi.org/10.35193/bseufbd.1075980    

---------------------------------------------

*The article will be ready for publication after the specified revisions are made.

**After revision, I would like to review the article again.

------------------------------------------------------

Congratulations to the authors.

I wish the authors success in their future academic studies.

Kind regards.

Author Response

Comment 1: The novelty of the study should be further explained (in introduction…).

Response: The authors agree with your suggestion. The novelty of the present study was clarified more widely.

Comment 2: More references (related to different studies) should be added to the introduction. (recent studies, 2021-2022).

Response: Thank you for suggestion. The references were revised.

Comment 3: In the introduction section; it will be useful to give general information about welding technologies.

Response: The authors agree with your suggestion. A brief explanation about widely used RSW was mentioned.

Comment 4: How were the experimental parameters chosen? (according to literature or?)

Response: Thank you for noticing this. The parameters were chosen by the preliminary experiments conducted to determine the range of used parameters. This information was also mentioned in the text.

Comment 5: The resolution of Figure 1, 3, 4 and 5 should be increased. (and magnify)

Response: The authors appreciate the suggestion. However, the presented figures were done with the high resolution of 300 ppi. Figures 1, 4 and 5 are magnified. The figure 3 shows the upper part of the welds and a corresponding cross section. The magnification of the cross section was explained and showed in the subsequent figures.

Comment 6: In «Results and discussion» section: …On the contrary, the material close to the interface between AA2024 and AISI304 plates did not experience any plastic deformation. The middle part of AA2024 plate had a very narrow zone with non-recrystallized and elongated grains (TMAZ)… Why? Explain? (more detailed)

Response: Thank you for pointing this out. The authors made a detailed explanation on microstructure formation on text. The effect of severe plastic deformation was seen in the upper part of the AA2024 plate which undergoes an elevated temperatures and rapid recrystallization. The middle part of AA2024 plate had a very narrow zone with non-recrystallized and elongated grains (TMAZ), caused by material flow and shear stress. The shear stress took place only in the upper and middle part of the aluminum plate, while the material close to the interface between AA2024 and AISI304 plates did not experience any plastic flow. This zone was influenced only by the heat. Furthermore, a slight thermal cycle did not permit the recrystallization of the aluminum. This was probably due to the lower conductivity of the steel and the presence of the Thermal Contact Conductance (TCC) at the interface, that represented the fraction of heat that is channeled from aluminum plate and the steel one. The TCC didn’t allow the complete heat exchange between the two plates and a fraction of heat persist in the AA2024 plate, decreasing the cooling rate.

- The graph in Figure 10 should be in more prominent colors.

Response: The authors thanks reviewer for the suggestion. The Graph was improved.

- The explanations about the hardness values given in Figure 10 should be detailed.

Response: Thank you for pointing this out. The authors made a detailed explanation on microhardness were implemented in the text. The upper area of aluminum plate was characterized with a higher microhardness with respect to the base metal due to great grain refinement and reprecipitation process that might take place during the cooling of heat-treatable AA2024 alloy, while next TMAZ and HAZ had a low one due to the coarsening or the dissolution of the hardening precipitates. The main differences in microhardness trend of samples AS-03 and AS-07 matched the extend material flow with high plastic deformation, where the most uniform microhardness distribution was in sample AS-07.

Comment 7: Conclusions section should be enriched a little more.

Response: Thank you for this suggestion. We agree that «Conclusion» section should cover more results. The text was modified.

Comment 8: More literature studies should be added to the introduction and other sections (DOIs given below).

Response: Proposed papers are powerful and fully related to the present study, thank you. The papers were added to the References.

Reviewer 4 Report

This is a timely effort by authors on "Modelling of Probe less Friction Stir Spot Welding of AA2024/AISI 304 steel lap joint". However, there are few suggestions to improve this manuscript further to make it look like a quality international paper;

1. The thermal properties table may be included for both grades of aluminium and steel.

2. There is a huge difference between the weld temperatures from experiments and numerical study. And numerical study gives temperature more than the melting point of aluminium. As FSSW is solid state method of welding metals, aluminium will be melting at the predicted temperatures. Then what is the vitality of numerical study towards thermal aspect of this research.

3. Equations 1 and 2 must be explained against the parameters in them.

4. For Figure 6, it is very difficult to recognize where the tool is actually interacting with the workpiece. The tool and workpiece should be clearly specified with arrows in the figures throughout this manuscript.

5. The lap shear joint must be shown with its complete geometrical dimensions.

6. Overall, the novelty seems to be little.

Author Response

Comment 1: The thermal properties table may be included for both grades of aluminium and steel.

Response: We agree with your suggestion. Table 2 «Thermal properties of materials» was included.

Comment 2: There is a huge difference between the weld temperatures from experiments and numerical study. And numerical study gives temperature more than the melting point of aluminium. As FSSW is solid state method of welding metals, aluminium will be melting at the predicted temperatures. Then what is the vitality of numerical study towards thermal aspect of this research.

Response: When 2 materials are in contact, the temperature at the bonding region arises for the presence of Thermal Contact Conductance/Thermal Contact Resistance. At the bonding there is a temperature drop between materials. The aluminum has a higher conductivity respects steel, and so the temperature arises. This is a local and circumscribed phenomenon, with a very low involved region, that could improve the join between aluminum and steel.

Comment 3: Equations 1 and 2 must be explained against the parameters in them.

Response: Thank you for noticing this. We have added the meaning of all variables in the text.

Comment 4: For Figure 6, it is very difficult to recognize where the tool is actually interacting with the workpiece. The tool and workpiece should be clearly specified with arrows in the figures throughout this manuscript.

Response: We agree that it would be better to mention the tool. Figure 6 was modified.

Comment 5: The lap shear joint must be shown with its complete geometrical dimensions.

Response: Thank you for this suggestion. The reason to present a part of cross-section with symmetry axis (Figure 3 b) is to combine them with the top view (Figure 3 a) in one figure while keeping the quality of both. The magnification of the cross section was explained and showed in the subsequent figures.

Comment 6: Overall, the novelty seems to be little.

Response: The novelty was clarified more widely. At now, there is very few studies on P-FSSW process on dissimilar (aluminum/steel) joint.

Reviewer 5 Report

Modelling of Probeless Friction Stir Spot Welding of AA2024/AISI 304 steel lap joint (2014230) – Materials

Reviewer's notes: 

This paper investigates the effect of rotation speed and tool force on the microstructure and microhardness of spot dissimilar welds in lap configuration made using a probeless tool. The experimental results and numerical results are compared in order to clarify temperature distribution and material flow. The subject of investigation is interesting and up to date. Here are the comments for authors: 

- Authors should give information on how the information in the Table 1 was obtained.

- Information in introduction overview about mismatch configurations in welding could be expanded and also include reference  https://doi.org/10.3390/ma15010214 

Author Response

We agree that mismatch configurations in welding is an important aspect. However, such observation is beyond the scope of our paper, which aims and considers only lap configuration. We have added the suggested paper to the manuscript on «Introduction» section. The chemical composition given in Table 1 was taken from the commercial documentation on the materials.

Round 2

Reviewer 3 Report

Manuscript ID Materials-2014230 entitled " Modelling of Probeless Friction Stir Spot Welding of AA2024/AISI 304 steel lap joint" for journal of Materials has been reviewed.

The authors have revised the manuscript carefully and the revised version could be published in the journal.

Decision- Accept

----------------------------------------------------------------------------

Congratulations to the authors.

I wish the authors success in their future academic studies.

Kind regards.